# Learning by Causality to Improve Channel Dependency Modeling in Multivariate Time Series Forecasting

## Abstract

Beyond the conventional long-term temporal dependency modeling, multivariate time series (MTS) forecasting has rapidly shifted toward channel dependency (CD) modeling. This shift significantly improves modeling quality by fully leveraging both multivariate relationships and temporal dependencies. Recent methods primarily model channel dependency through correlation learning (e.g., cross-attention) or non-trainable statistical techniques (e.g., cross-correlation), however, these approaches are insufficient to fully capture the intrinsic relationships within MTS, particularly those stemming from directed cause-effect (i.e., causality) and non-stationary variates originating from diverse sources. In addition, causality may arise in MTS with different temporal behaviors, such as varying periodicity or discrete event sequences, which remains underexplored. In this paper, we propose CALAS (Causality-enhanced Attention with Learnable and Adaptive Spacing), the first end-to-end learning method for MTS forecasting that uncovers causality among variates without relying on statistical measures or prior knowledge. To model underlying causality, which consists of causal strength and propagation delay, we newly design a hypernetworks-based 1D convolutions mechanism. Inspired by dilated convolution with learnable spacings (DCLS) and spiking neural networks (SNNs), we extend discrete time delay into a continuous Gaussian kernel. Combining the hypernetworks-generated Gaussian kernel and convolutional weights (i.e., attention or causal strength), we achieve the end-to-end dynamic causality modeling mechanism, improving the prediction accuracy, quality, and interpretability of both MTS forecasting results and models. For evaluation, we conduct extensive experiments with six real-world datasets and qualitative analysis to demonstrate CALAS's superiority in capturing varying causality in a data-agnostic manner. The experiment results indicate that CALAS has significantly improved MTS forecasting accuracy compared to state-of-the-art methods by dynamically modeling causality among variates.

## 1 Introduction

Multivariate time series (MTS) forecasting is one of the core problems in time series analysis, widespread in various domains such as traffic (Li & Shahabi, 2018; Lee et al., 2020; Wu et al., 2020), finance (Zeng et al., 2023; Zhu & Shasha, 2002), and weather (Matsubara et al., 2014; Wu et al., 2021). The fundamental idea of MTS forecasting methods is to build a mapping function from input (i.e., historical) to output (i.e., future) MTS by learning inherent dependencies of MTS. To model the intrinsic dependencies of MTS, the main flow of MTS forecasting is evolved into two distinguished categories – temporal dependency modeling and channel (or, variate) dependency modeling.

For temporal dependency modeling, researchers have proposed novel deep learning architectures, ranging from RNN-based models (Li et al., 2018; Jiang et al., 2023; Zonoozi et al., 2018) to Transformer-based architectures (Vaswani et al., 2017; Zhou et al., 2021; Liu et al., 2022), or have modified existing model for improved dependency modeling (Bai et al., 2020; Wu et al., 2021; Zhou et al., 2022). While these advances have significantly improved MTS forecasting quality, they often fail to capture important temporal characteristics, such as periodicity (Oreshkin et al., 2020; Wu et al., 2021; Zeng et al., 2023). To address this issue, some approaches incorporate prior knowledge

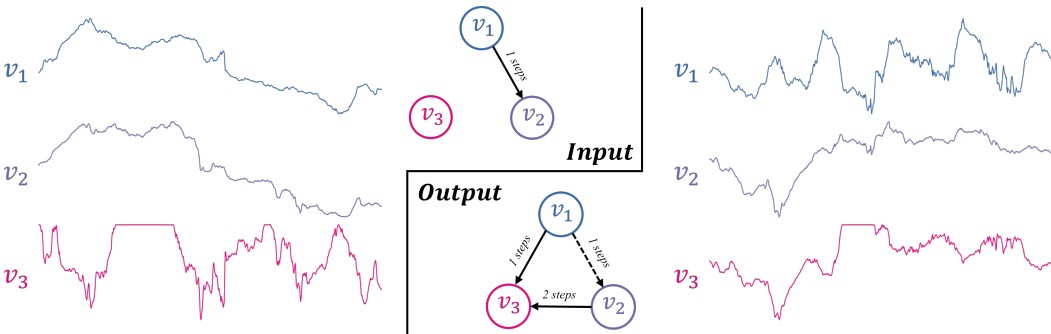

Figure 1: Distribution shift over lead-lag relationships in the Weather dataset (1,3, and 4-th variates in the dataset). (Left) In the historical data from time $t - T + 1$ to $t - 1$, two variates, $v_1$ and $v_2$, exhibit a strong causality, while $v_3$ has no causal relation with $v_1$ and $v_2$ (top causal graph). (Right) In the consecutive output signals, the relationship between $v_1$ and $v_2$ weakens, while both $v_1$ and $v_2$ newly establish a strong causal relation with $v_3$, invoking need for a layer-agnostic causality modeling.

from time series analysis, such as time series decomposition (Wu et al., 2021; Zeng et al., 2023; Zhou et al., 2022; Nie et al., 2023), or integrate statistical methods, including correlation and Fourier transformation (Zhou et al., 2022; Wu et al., 2023; Yi et al., 2023b), into their models. Nie et al. (2023) have recently introduced a patch-based input representation that enables models to learn semantic information directly without prior knowledge. Additionally, several methods have been developed to handle non-stationary behavior and distribution shifts frequently observed in real-world MTS data (Kim et al., 2022; Zhou et al., 2023).

On the other hand, a line of research has focused on modeling channel dependencies (CDs) in MTS, named channel dependency (CD) modeling. Their strategies can be categorized into two main approaches: correlation-based and causality-based. In correlation-based methods, CD are typically captured using advanced neural architectures, such as GNNs (Guo et al., 2019; Wu et al., 2019; Li et al., 2018), CNNs (Wu et al., 2023; Zeng et al., 2024; donghao & wang xue, 2024), and Attention mechanisms (Zhang & Yan, 2023; Liu et al., 2023; Zhou et al., 2021; Wang et al., 2024), often incorporating Fourier transformation and multi-periodicity decomposition (Wu et al., 2023; Yi et al., 2023a). Recently, some researchers have argued that CDs in MTS are inherently directional, meaning they involve cause-and-effect relationships (i.e., causality) and cannot be fully understood through correlation-based, non-directional models (Wu et al., 2020; Li & Shahabi, 2018; Zhao & Shen, 2024; Lee et al., 2022). Certain GNN-based methods attempt to learn cause-effect relationships using node embeddings; however, they struggle to fully capture the dynamic nature of causality (Wu et al., 2020). Recently, Zhao & Shen (2024) have proposed a novel method that leverages causality extracted from cross-correlation to align output signals with propagation delay along with corresponding leading indicators. While this approach addresses dynamic changes in causality, questions remain about the capacity of cross-correlation and sensitivity to hyperparameters (e.g., the maximum number of leading indicators). These issues highlights two ongoing challenges in adopting causality modeling in MTS: 1) lack of end-to-end, learning-based dynamic causality modeling methods; and 2) limitation of prior knowledge or statistical methods in fully leveraging intrinsic causality in MTS.

Figure 1 presents a motivating example from the Weather dataset, where local causality changes in consecutive signals, highlighting the need for dynamic causality modeling. It also emphasizes a need for layer-agnostic causality modeling methods that can be applied in any level of input, intermediate, or output layers, since causal structure may changed even within model forwarding. Additionally, it suggests a potential side-effect of statistic-based CD modeling approaches, such as cross-correlation (e.g., LIFT (Zhao & Shen, 2024)), which could mislead models to focus on less relevant variates.

Moreover, correlation-based methods often struggle with long-term MTS, especially in signals with strong short-term periodicity. To further describe this phenomena, we present Figure 2 that shows synthetic signals with a periodic seasonal pattern, sinusoidal trends, and varying noise level. While all three signals shares the same seasonal pattern, $v_2$ and $v_3$ exhibit a stronger causal relationship than others. Nevertheless, statistical methods tend to emphasize the seasonal patterns, leading to errors in identifying the correct lead-lag relationship, as illustrated in the bottom graph of Figure 2.

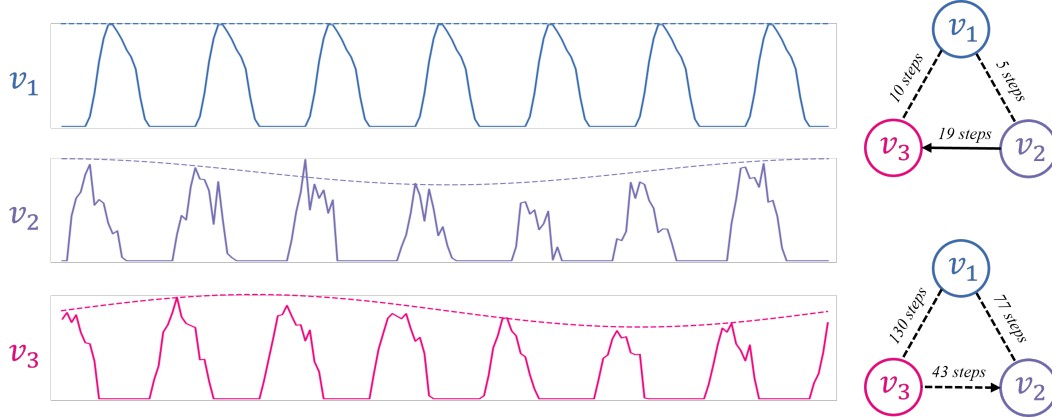

Figure 2: A illustration of long-term lead-lag relationships over the synthetic signal with seasonality and trend. Dashed line in time series shows trends. The top-right graph is the actual causality graph during data generation, while the bottom-right one is causality graph calculated with cross-correlation. In causality graph, dashed line means weak (or possibly undirected) relationship. There are large misalignment on the lead-lag steps.

In this work, we propose a novel end-to-end causality modeling method named CALAS[1] (Causality-enhanced Attention with Learnable and Adaptive Spacing) that converts correlation-based attention into causality by incorporating dilated convolution with learnable spacing (DCLS) (Hassani et al., 2023; Khalfaoui-Hassani et al., 2023) and hypernetworks (Ha et al., 2017) in a novel way. CALAS consists of three components: two hypernetworks and one convolutional neural network. To uncover the dynamic causal relations among MTS while considering its propagation delay, we extend the static delay modeling with the Gaussian kernel by introducing input-based dynamic delay modeling. Specifically, we adopt the hypernetworks–neural networks that generate the weights for another network–to model both the dynamic causal strength and propagation delay (i.e., causality map). We utilize two hypernetworks: one for estimating the Gaussian kernel, which serves as the time lag (or delay) of causality, and the other to measure the strength of the causality. After generating the weights, CALAS applies the resulting causality map directly through the convolutional architecture. In addition to enabling flexible causality learning beyond the methods relying on prior knowledge, our design enhances interpretability by separating causality map into causal strength and propagation delay. Furthermore, unlike the other causality-based CD models, CALAS can learn causality in input space, output space, or both, offering greater flexibility in integrating CALAS with other causality modeling methods, which will be discussed in the experimental results.

The main contributions of this paper can be summarized as follows.

- We introduce a new perspective on converting the correlation-based attention score into causality-based relationships by decomposing the causality map into learnable delay and causal strength.

- We design a novel architecture, CALAS, that enables the end-to-end learning of dynamic cause-and-effect relationships among variates for MTS forecasting. By design, CALAS can be applied in a model-agnostic way at any stage of the MTS forecasting models, including input and output space. To the best of our knowledge, CALAS is the first work to model causality for MTS forecasting in an end-to-end manner.

- Experimentally, we demonstrate that CALAS significantly improves the state-of-the-art methods. Specifically, our model improves the existing models by 7.61% on average, outperforming existing methods (e.g., LIFT) by 2.21 percentage points. We highlight that the improvements become larger for long-term forecasting, 9.45% on average. We have also revealed that CALAS can be integrated with other causality-based plug-and-play modules, such as LIFT, resulting in substantial performance gains.

---

[1]Github link for source code will be provided.

## 2 RELATED WORK

MTS forecasting could be categorized into two research theme: temporal dependency modeling without inter-variate dependencies (i.e., channel-independent methods) and channel dependency (CD) modeling methods. Since our paper aims to solve CD modeling problem, we briefly introduce CI methods and move on to CD methods.

In MTS forecasting, CI methods are well-studied and are still on-going research directions, ranging from statistical models (e.g., ARIMA (Box et al., 2015), hidden Markov model (Pesaran et al., 2004)) to recent deep learning models (Liu et al., 2023; Zhao & Shen, 2024). Recent models are focusing on the long-term dependency modeling by leveraging the traditional studies, such as Fourier transformation (Zhou et al., 2022; Yi et al., 2023b; Xu et al., 2024) or by proposing novel input representation methods (Nie et al., 2023; Wu et al., 2021; Zeng et al., 2023; Hu et al., 2024; Jia et al., 2023). Those methods contribute to MTS forecasting domain by deeply understanding in-nature characteristics of time series (i.e., temporal dependency).

On the other hands, CD methods aim to uncover the intrinsic channel-wise dependency. Based on the methodology and assumption for channel dependency, CD methods can be divided into four categories: 1) model-based dynamic correlation learning (Learnable Correlation in Table 1), 2) model-based static causality learning (Learnable Static Causality in Table 1), 3) statistics-based dynamic causality learning (Statistics-based Causality in Table 1), and 4) model-based dynamic causality learning (Ours in Table 1). For CD modeling, there are primarily two directions–learnable correlation with attention mechanism (Zheng et al., 2020; Vaswani et al., 2017; Behrouz et al., 2024; Zhang & Yan, 2023) and learnable static causality with graph neural networks (GNNs) and learnable embedding (Cao et al., 2020; Guo et al., 2019; Wu et al., 2020; 2019; Lee et al., 2022). While attention mechanism aims to reconstruct correlation map by projecting inputs into the feature space, GNN-based methods try to approximate underlying, time-independent causal structure with learnable node embedding. However, those approaches merely discuss the lead-lag relationship and propagation delay, which is crucial for the CD modeling.

Recently, a few researches (Wu et al., 2020; Zhao & Shen, 2024) depict that when two signals have cause-effect relationship, there exist a time-lag for 'cause' signal to influence the 'effect' signal, which we call propagation delay. To fully leverage the benefits of CD methods, it is necessary to consider propagation delay, since ignorance of propagation delay making the model refer the outdated information (Zhao & Shen, 2024). To the best of our knowledge, LIFT (Zhao & Shen, 2024) is the only work that explicitly aims to solve the propagation delay by leveraging cross-correlation, which is statistics-based causality in our categorization. However, overrelying the statistical measure introduces a number of limitations, such as the information bottleneck from the excessively large number of recurring patterns (Lee et al., 2022), possible misalignment, and need of reference signals that limit the utilization of model in various situation. We describe those properties in Table 1.

Table 1: Comparison of existing CD methods. $\Delta$ means the method partially satisfies the condition.

| Methods | Learnable Correlation | Learnable Static Causality | Statistics-based Causality | Ours (CALAS) |
|---|---|---|---|---|
| Delay-awareness | × | × | ✓ | ✓ |
| Directed | × | ✓ | $\Delta$ | ✓ |
| Dynamic | ✓ | × | ✓ | ✓ |
| Learnable | ✓ | ✓ | × | ✓ |
| Example Methods | Attention (Vaswani et al., 2017) | GNNs (Wu et al., 2020) | LIFT (Zhao & Shen, 2024) | None (Ours) |

## 3 PRELIMINARIES

Let $\mathbf{x}_t \in \mathbb{R}^N$ denote a multivariate time series (MTS) with $N$ variates (i.e., channels) at a time step $t$. Given $T$-length historical observation $\mathbf{X}_t = \{\mathbf{x}_{t-T+1}, \ldots, \mathbf{x}_t\} \in \mathbb{R}^{T \times N}$ at time $t$, MTS forecasting aims to predict corresponding future $L$-length MTS $\mathbf{Y}_t = \{\mathbf{x}_t, \ldots, \mathbf{x}_{t+L}\} \in \mathbb{R}^{L \times N}$. In the remaining sections, we denote the $i$-th variate of MTS as $\mathbf{X}^{(i)} \in \mathbb{R}^T$, and each point at time $t$ as $\mathbf{x}_t^{(i)}$. Formally, MTS forecasting could be described as below:

**Definition 3.1.** Given MTS data $\mathbf{X}_t$, MTS forecasting aims to approximate the model $f(\cdot, \theta) : \mathbb{R}^{T \times N} \rightarrow \mathbb{R}^{L \times N}$ by minimizing $L(f(\mathbf{X}_t, \theta), \mathbf{Y}_t)$.

For the channel-independent (CI) models, $f(\cdot, \theta)$ shares the same parameter for all variates, enabling the universal representation learning for MTS, on the other hand, it merely models the channel dependency that is crucial in MTS forecasting. It therefore focuses more on the temporal dependency modeling.

Given MTS $\mathbf{X}_t$ at time $t$, we assume that there is unknown dynamic causal graph $\mathbf{A}_t$, where each element (i.e., relationship from $i$-th variate to $j$-th variate) is defined as $a_{i,j} = \{w_{i,j}, \delta_{i,j}\} \in \mathbb{R} \times \mathbb{N}$. $w_{i,j}$ means the strength of causality and $\delta_{i,j}$ means the discrete delay. In addition to the temporal dependency, Channel dependency (CD) modeling methods aim to approximate the causal graph $\mathbf{A}_t$ as below:

**Definition 3.2.** Channel dependency modeling aims to approximate $f(\cdot, \theta | \tilde{A}_t)$ that minimize $L(f(\mathbf{X}_t, \theta | \tilde{A}_t), \mathbf{Y})$, where $\tilde{A}_t$ is approximation of causal graph $\mathbf{A}_t$ with learnable parameter $\theta_A$ such that $\tilde{A}_t = g(\theta_A)$ or $\tilde{A}_t = g(\mathbf{X}, \theta_A)$.

### 3.1 DETAILED CATEGORIZATION FOR CHANNEL DEPENDENCY MODELING

From the literature review, we have identified two distinctions in CD methods: the perspective on temporal variation and the type of relationship. Regarding temporal variation, CD modeling can be classified as either static or dynamic. Static CD modeling assumes that the underlying causal structure remains constant over time (i.e., $\mathbf{A}_t = \mathbf{A}_{t'} \forall t, t'$), thus avoiding instantaneous effects or potential misleading caused by anomalous values. This approach is widely used in early traffic forecasting (Wu et al., 2020; 2019; Li et al., 2018; Lee et al., 2022) and causal discovery (Cheng et al., 2023b; 2024). In contrast, dynamic CD modeling assumes that CD can change over time. It is a dominant research trend in long-term MTS forecasting, including Crossformer (Zhang & Yan, 2023) and LIFT (Zhao & Shen, 2024). Dynamic CD modeling offers the advantage of adjusting the strength or nature of dependencies and also help mitigate distribution shifts in channel dependency (Zhao & Shen, 2024; Jiang et al., 2023).

Additionally, CD can be categorized into two types of relationships: undirected and directed. Correlation, an undirected relationship, is a traditional approach to modeling time series data. Transformer-based models are a representative example, utilizing self-attention mechanism to model channel dependencies. Recent patch-based models (Nie et al., 2023; Zhang & Yan, 2023) or inverted embedding methods (Liu et al., 2023) also fall under correlation-based model due to their input representations. On the other hand, there are a few researches focus on directed causal relation, using GNNs (Wu et al., 2020; Jiang et al., 2023) or statistical methods (Zhao & Shen, 2024). Directed CD methods explicitly target lead-lag, propagation, or causality as their objective, often relying on either static structure learning (Wu et al., 2020) or statistical methods, such as cross-correlation (Zhao & Shen, 2024). Moreover, there is no literature that explicitly learns the delay in MTS, limiting the interpretability.

## 4 METHODS

In this section, we introduce proposed methods, CALAS, a end-to-end dynamic causality modeling methods for MTS forecasting. Before we detail CALAS, we revisit the Granger causality and causal discovery in real-world MTS data to discuss the gap between the real-world MTS and causal discovery. We then explain decomposition of the causality graph into strength and propagation delay. It disentangles the causal strength and time-lag for propagation, enabling in-depth analysis over channel dependencies. Afterwards, we explain the propagation of causal effects in terms of 1D convolution (Section 4.2). Lastly, we describe how CALAS achieves dynamic causality modeling with two hypernetworks (Section 4.3). Our model first generate the convolutional weights (Section 4.3) and then feed them into Gaussian kernel (Equation 4 in Section 4.2). Then, it applies generated causal graph with 1D convolution (Equation 3) and forward to next layer (in case of Figure 3a).

**Motivations** Even though many researches discuss the importance of **propagation delay** (Wu et al., 2020; Zhao & Shen, 2024; Jin et al., 2023), which means the time taken for a cause signal to actually influence the effect signal, there are less effort to explicitly model the discrete delay in MTS forecasting. This ignorance eventually induces the misleading of causality by introducing additional complexity in CD modeling. Recent research utilizes cross-correlation to find the lead-lag relationship and its propagation delay, however, it also relies on the causal strength rather than actually model

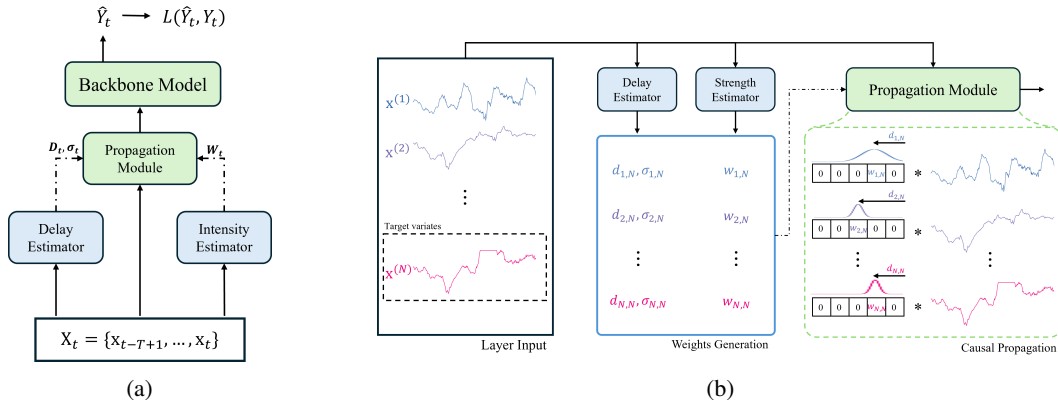

Figure 3: (a) CALAS when applied to input space and (b) an overview of CALAS forwarding path with respect to output channel $N$. Given input MTS (b, left), two hypernetworks of delay and causal strength estimators calculate propagation delay, standard deviation, and causal strength (b, middle). Standard deviation is utilized to convert discrete delay into continuous form. Finally, Gaussian 1D convolutional kernel is constructed and CALAS operates causal propagation with the propagation module (b, right). For the sake of simplicity, we assume that $\sigma$ is sufficiently small, so we have an almost discrete kernel. The two blue modules are hypernetworks and dash-dotted line means supplying the generated parameters from the hypernetworks.

the discrete delay (Zhao & Shen, 2024). The importance of delay learning is also actively discussed and is demonstrated in spiking neural networks, which also solves the problem with propagation delays (Maass, 1997; Maass & Schmitt, 1999; Shrestha & Orchard, 2018; Hammouamri et al., 2024). Maass & Schmitt (1999) theoretically and experimentally demonstrated that $k$ adjustable delays can compute a much richer class of functions than one with $k$ adjustable weights. Motivated by those factors, we newly introduce hypernetworks-based 1D convolution mechanism that learns both dynamic causal strength and propagation delay in end-to-end manner.

## 4.1 CAUSALITY, GRANGER-CAUSES, AND CAUSAL DISCOVERY IN REAL-WORLD MTS DATA

**Causality**   In the causal discovery domain, the causality is commonly based on several assumptions: Markovian condition, faithfulness, sufficiency, no instantaneous effect, and stationarity (Cheng et al., 2024). However, in the real-world MTS, it is hard to observe "all possible events and causes in the world," the sufficiency–all common causes of variables are observed–and stationarity–all the causal relationships remain constant throughout time– are hard to be hold. For the real-world MTS data, we need to assume that there exists latent variables or events that changes causality of observed variates. Therefore, in this paper, *causality* is based on the assumptions for Markovian condition, faithfulness, and sufficiency. We emphasize that *causality* in this paper is not necessarily be **true causality**, which have assumptions for no latent compounder and no unseen events.

**Concepts of CALAS**   The basic idea of CALAS stems from the concepts of multivariate Granger causality (GC) analysis (Granger, 1969). Multivariate GC analysis is performed by fitting a VAR model with given maximum possible propagation delay $k$ as below:

$$X_t = \sum_{\tau=1}^{k} W_\tau X_{t-\tau} + \epsilon(t), \tag{1}$$

where $W_\tau \in \mathbb{R}^{N \times N}$ is an estimated causal strength with propagation delay $\tau$. Several approaches are proposed to resolve limitations on GC, such inability to capture non-linear causal relationship (Tank et al., 2022; Cheng et al., 2023b). However, they stick to the static and individual causality measure for each time lag, disregarding the continuous property of MTS. To resolve these limitations, we set our design rationales as: 1) time lag (i.e., propagation delay) for two time series should be unique and be estimated with continuous function, 2) causal strength and propagation delay should be disentangled, and 3) the causality estimation should be time-dependent, more precisely, should handle latent compounder and unseen events.

## 4.2 Causal Propagation into Temporal Convolution

Given $\mathbf{X}^{(i)}$ and $\mathbf{X}^{(j)}$, let assume that they have $w_{i,j}$ lead-lag relationship with $d_{i,j}$ propagation delay. Let assume that we can derive $\mathbf{x}_t^{(i)}$ from a set of 'cause' signals. Then, we can formulate $\mathbf{x}_t^{(i)}$ as below:

$$\mathbf{x}_t^{(i)} = \sum_{j \in \mathcal{N}_t^{(i)}} w_{i,j} \mathbf{x}_{t-d_{i,j}}^{(j)}, \tag{2}$$

where $\mathcal{N}_t^{(i)}$ is a set of 'cause' signals with respect to $\mathbf{x}_t^{(i)}$ at time $t$. In our paper, we assume that all variates could act as potential cause signals (i.e., $|\mathcal{N}_t^{(i)}| = N$) at model initialization. We can reformulate Equation 2 into one dimensional temporal convolution with Dirac delta function $\delta$ as follows:

$$\mathbf{x}_t^{(i)} = \sum_{j \in \mathcal{N}_t^{(i)}} w_{i,j} \delta(t - d_{i,j}) * \mathbf{x}_t^{(i)}, \tag{3}$$

where $*$ is 1D convolution and $\delta(t - a) * f(t) = f(t - a)$. We have detailed this process in right side of Figure 3b. To approximate Dirac delta function, we utilize Gaussian kernel with maximum delay (kernel size in trivial 1D convolution) (Hassani et al., 2023; Khalfaoui-Hassani et al., 2023). Let assume that we have $k$-size convolutional kernel for our model. Then, we approximate Gaussian kernel centered at $k - d_{i,j} - 1$, where $d_{i,j} \in [0, k-1]$, with standard deviation $\sigma_{i,j} \in \mathbb{R}^*$. Then, $\forall n \in [0, \ldots, k-1]$ our $k$-size convolutional kernel will be:

$$k_{i,j}[n] = \frac{w_{i,j}}{A} exp\left( -\frac{1}{2} \left( \frac{n - k + d_{i,j} + 1}{\sigma_{i,j}} \right)^2 \right), \text{where} \tag{4}$$

$$A = \epsilon + \sum_{n=0}^{k-1} exp\left( -\frac{1}{2} \left( \frac{n - k + d_{i,j} + 1}{\sigma_{i,j}} \right)^2 \right)$$

with $\epsilon \in \mathbb{R}^*$ to avoid division by zero. In Equation 4, $w_{i,j} \in \mathbb{R}$, $d_{i,j} \in [0, k-1]$, and $\sigma \in \mathbb{R}^*$ are trainable parameter. Gaussian kernel converts the discrete Dirac delta function, which is not differentiable, into the continuous function, enabling the gradient calculation. To avoid underfitting of the delay, we penalize the $\sigma$ with boundary value $v$, which continuously decreases up to 0.01 during the training phase. In the inference phase, we replace the Gaussian kernel with pulse function (i.e., hard voting). As a results, we can obtain almost discrete kernel representation as Figure 3b.

Despite the Gaussian kernel-based 1D convolution successfully parameterized both propagation delay and causal strength, it still has two limitations for dynamic causality modeling. First of all, trained causal map is fixed matrix and will not changed. It thus will be vulnerable to distribution shifts of causality. Secondly, it cannot directly observe and utilize the input information and lead-lag relationship, which make the model hardly understand intrinsic nature of MTS. Therefore, we additionally introduce hypernetworks to generate parameters in Equation 4.

## 4.3 Causality-enhanced Attention with Learnable and Adaptive Spacing

In this section, we describe the overall flow of CALAS with hypernetworks when it applied to input layer, as illustrated in Figure 3a. Specifically, we utilize two hypernetworks, one for the causal strength, and the other for delay learning. The generation process are shared for both hypernetworks, however, the causal strength is shared for k possible delays. Specifically, in the Equation 4, $w_{i,j} \in \mathbb{R}, \sigma_{i,j} \in \mathbb{R}, d_{i,j} \in \mathbb{R}^k$. By sharing same causal strength for the same pair of variates, CALAS achieves better generalization for causal strength.

We design our hypernetworks with one multi-layer perceptron (MLP) and feature embedding. MLP layer takes the input vector $z \in \mathbb{R}^{T \times N_i}$ (in Figure 3a, $\mathbf{X}_t$), where $N_i$ is input channels, and project it into hidden size of $H$. To map the projected input to the $N_o$ channels, we introduce embedding vectors $\mathbf{M} = \{\mathbf{m}^{(1)}, \ldots, \mathbf{m}^{(N_o)}\}$, where $\mathbf{m}^{(i)} \in \mathbb{R}^{h \times N_o}$. $\mathbf{M}$ is initially random vectors and will be trained to represent the static features of each output variate. For the output channels, we can

interchangeably use MLP or embedding vectors with users' intention. Thus, we generate weights $w_{i,j}$ as follows:

$$w_i^{in} = w_{MLP}z^{(i)} + b_{MLP}; w_{i,j} = g(w_i^{in}, \mathbf{m}^{(j)})$$

where $w_{MLP} \in \mathbb{R}^{T \times H}$ and $b_{MLP} \in \mathbb{R}^H$ are learnable parameters. Function $g(\cdot, \cdot)$ could be any function or operation to generate $\mathbb{R}^{N_i \times N_o}$ map, including matrix multiplication and self-attention. To generate causal strength, we utilize matrix multiplication with LeakyReLU activation function, resulting to $\mathbb{R}^{N_i \times N_o}$ vector. For the propagation delay, we mimic multi-head attention (Vaswani et al., 2017), which generates $\mathbb{R}^{N_i \times N_o \times k}$ output. Generated weights will be directly feed into Gaussian kernel (Equation 4) and then forward to propagation module (Figure 3b). CALAS will be served as the plug-in module injected in input-, output-, or intermediate-layers. In case of Figure 3a, CALAS has plugged into the input layer and its output will be served as the input for backbone model.

## 5 EXPERIMENTS

### 5.1 EXPERIMENTAL SETTINGS

**Datasets**   Following previous researches (Zhou et al., 2021; 2022; Nie et al., 2023), we conduct experiments on six MTS datasets, including Weather, Electricity, Traffic, Solar, Wind, and PeMSD8. In this paper, we have excluded ETT* datasets because they are recorded from the same sources, which means they simultaneously changes and have no causal relationship (Zhou et al., 2021). We also exclude Exchange-rate dataset since it is well known that $x_{t-1}$ is the best predictor for $x_t$ if the market is efficient (Fama, 1970; Rossi, 2013) and Zeng et al. (2023) have experimentally demonstrated that simply repeating the last value can outperform or be comparable to the best results.

**Baselines**   To demonstrate performance of CALAS as a plug-in module, we conduct the experiments with CALAS and four MTS forecasting backbones, PatchTST (Nie et al., 2023), DLinear (Zeng et al., 2023), Crossformer (Zhang & Yan, 2023), and MTGNN (Wu et al., 2020). PatchTST and DLinear is channel-independent modeling methods and others are channel-dependent methods. We further compared our results with statistics-based plug-in module, LIFT (Zhao & Shen, 2024), to prove effectiveness of end-to-end dynamic causality modeling. We also provide comparative experiments with Linear model-based CALAS with baseline methods, additionally including FEDformer (Zhou et al., 2022), Autoformer (Wu et al., 2021), and LightMTS (Zhao & Shen, 2024).

**Setups**   We follow the same experimental settings as previous MTS forecasting works (Zhou et al., 2022; Nie et al., 2023), where the forecasting horizon $L \in \{96, 192, 336, 720\}$ and lookback window $T = 336$. All the baselines follow the experimental settings, including hyperparemeter settings, in official code. For CALAS, we adopt the Adam optimizer and search the optimal learning rate in $\{0.1, 0.05, 0.01, 0.005, 0.001, 0.0005\}$. We set hidden size $H = 64$ and batch size to 32. We have run each experiment five times and reported average performance. We also provide quantitative and qualitative results for causal discovery in Appendix A.

### 5.2 MAIN RESULTS

Table 2 indicates the experimental results among four state-of-art methods, LIFT, and CALAS on six MTS datasets. CALAS significantly improves the backbone models by 6.65% on average, outperforming LIFT by 2.01 percentage points. This results indicates both importance of end-to-end dynamic causality modeling and limitation of statistical methods, especially in long-term forecasting. We have omitted the error bars for the experimental results since LIFT and CALAS are stable and has less than $0.001$. We provide comparative analysis and depict important findings in following paragraphs.

**Improvements over backbone models**   CALAS significantly improve the performance of backbone models, implying the importance of both channel dependency and propagation delay. For channel-independent (CI) methods, such as PatchTST and DLinear, CALAS achieves an average improvements of 9.88%. We particularly highlight the Weather, Solar, and PeMSD8 datasets, which have shown the largest gains with CALAS. These results are attributed to the complex spatio-temporal dependencies

Table 2: Performance comparison in terms of forecasting errors. The **bold text** means the best results among backbone, with LIFT, and with CALAS and underlined text means the best results among all the methods for each dataset. We also provide improvements over 1) baseline and 2) LIFT in the second-rightmost and rightmost columns, respectively. Impr. over LIFT displays percent point of improvements.

| Method | PatchTST MSE | MAE | +LIFT MSE | MAE | +CALAS MSE | MAE | DLinear MSE | MAE | +LIFT MSE | MAE | +CALAS MSE | MAE | Crossformer MSE | MAE | +LIFT MSE | MAE | +CALAS MSE | MAE | MTGNN MSE | MAE | +LIFT MSE | MAE | +CALAS MSE | MAE | Impr. (baseline) | Impr. (LIFT) |
|---|---|---|---|---|---|---|---|---|---|---|---|---|---|---|---|---|---|---|---|---|---|---|---|---|---|---|
| **Weather** | | | | | | | | | | | | | | | | | | | | | | | | | | |
| 96 | 0.152 | 0.199 | 0.146 | 0.196 | **0.144** | **0.192** | 0.176 | 0.237 | **0.145** | 0.203 | 0.149 | **0.200** | 0.145 | 0.209 | 0.146 | 0.210 | **0.144** | 0.211 | 0.157 | 0.216 | 0.154 | 0.212 | **0.150** | **0.208** | 6.76 | 1.06 |
| 192 | 0.197 | 0.243 | **0.190** | 0.238 | 0.192 | **0.236** | 0.220 | 0.282 | 0.189 | 0.249 | **0.188** | **0.247** | 0.197 | 0.264 | 0.196 | 0.262 | **0.195** | **0.261** | 0.205 | 0.269 | 0.203 | 0.266 | **0.198** | **0.263** | 5.59 | 0.82 |
| 336 | 0.249 | 0.283 | 0.243 | 0.281 | **0.240** | **0.277** | 0.265 | 0.319 | 0.243 | 0.292 | **0.238** | **0.290** | 0.246 | 0.309 | 0.245 | 0.305 | **0.244** | **0.304** | 0.258 | 0.312 | 0.256 | 0.308 | **0.251** | **0.300** | 4.56 | 1.56 |
| 720 | 0.320 | 0.335 | 0.315 | 0.333 | **0.301** | **0.325** | 0.323 | 0.362 | 0.317 | 0.349 | **0.289** | **0.337** | 0.323 | 0.364 | 0.321 | 0.360 | **0.317** | **0.358** | 0.335 | 0.369 | 0.333 | 0.365 | **0.322** | **0.355** | 5.02 | 4.01 |
| **Electricity** | | | | | | | | | | | | | | | | | | | | | | | | | | |
| 96 | 0.130 | 0.222 | 0.128 | 0.222 | **0.127** | **0.212** | 0.140 | 0.237 | **0.130** | **0.225** | 0.131 | 0.226 | 0.142 | 0.243 | 0.138 | 0.238 | **0.137** | **0.236** | 0.138 | 0.238 | 0.133 | 0.233 | **0.131** | **0.231** | 4.23 | 1.15 |
| 192 | 0.148 | 0.240 | 0.147 | **0.239** | **0.144** | 0.241 | 0.153 | 0.249 | 0.148 | **0.242** | **0.147** | 0.245 | 0.159 | 0.259 | 0.154 | 0.251 | **0.152** | **0.247** | 0.160 | 0.261 | 0.153 | 0.252 | **0.152** | **0.249** | 3.45 | 0.79 |
| 336 | 0.167 | 0.261 | 0.163 | 0.257 | **0.162** | **0.246** | 0.169 | 0.267 | 0.163 | 0.261 | **0.161** | **0.257** | 0.192 | 0.293 | 0.176 | 0.276 | **0.173** | **0.272** | 0.193 | 0.284 | 0.187 | 0.275 | **0.184** | **0.268** | 5.95 | 2.20 |
| 720 | 0.202 | 0.291 | 0.195 | 0.289 | **0.190** | **0.282** | 0.203 | 0.301 | **0.190** | 0.295 | 0.193 | **0.294** | 0.264 | 0.353 | 0.224 | 0.312 | **0.213** | **0.311** | 0.242 | 0.327 | 0.216 | 0.305 | **0.213** | **0.295** | 9.87 | 2.76 |
| **Traffic** | | | | | | | | | | | | | | | | | | | | | | | | | | |
| 96 | 0.367 | 0.251 | 0.352 | **0.242** | **0.350** | 0.243 | 0.410 | 0.282 | 0.394 | 0.273 | **0.388** | **0.269** | 0.519 | 0.293 | 0.462 | 0.284 | **0.457** | **0.277** | 0.479 | 0.289 | 0.464 | 0.286 | **0.461** | **0.282** | 5.55 | 1.28 |
| 192 | 0.385 | 0.259 | 0.373 | **0.251** | **0.371** | **0.251** | 0.423 | 0.287 | 0.413 | 0.281 | **0.408** | **0.278** | 0.522 | 0.296 | 0.490 | 0.283 | **0.488** | **0.280** | 0.507 | 0.307 | 0.491 | 0.301 | **0.480** | **0.299** | 4.36 | 1.04 |
| 336 | 0.398 | 0.265 | 0.389 | 0.262 | **0.385** | **0.259** | 0.436 | 0.296 | 0.426 | 0.288 | **0.421** | **0.285** | 0.530 | 0.300 | 0.517 | 0.303 | **0.494** | **0.293** | 0.539 | 0.314 | 0.519 | 0.309 | **0.512** | **0.301** | 4.05 | 2.33 |
| 720 | 0.434 | 0.287 | 0.429 | 0.286 | **0.425** | **0.283** | 0.466 | 0.315 | 0.454 | 0.307 | **0.443** | **0.297** | 0.584 | 0.369 | 0.543 | 0.322 | **0.536** | **0.312** | 0.616 | 0.352 | 0.532 | 0.340 | **0.520** | **0.335** | 8.19 | 2.40 |
| **Solar** | | | | | | | | | | | | | | | | | | | | | | | | | | |
| 96 | 0.176 | 0.227 | **0.174** | **0.224** | 0.174 | 0.225 | 0.222 | 0.291 | 0.185 | 0.238 | **0.178** | **0.233** | 0.179 | 0.245 | 0.176 | 0.224 | **0.176** | **0.221** | 0.167 | 0.224 | 0.166 | 0.223 | **0.160** | **0.214** | 9.16 | 2.11 |
| 192 | 0.205 | 0.260 | 0.190 | 0.245 | **0.188** | **0.243** | 0.249 | 0.309 | 0.194 | 0.253 | **0.190** | **0.252** | 0.204 | 0.254 | 0.197 | 0.250 | **0.181** | **0.248** | 0.180 | 0.243 | 0.179 | 0.239 | **0.169** | **0.228** | 12.2 | 3.61 |
| 336 | 0.200 | 0.252 | 0.194 | 0.249 | **0.192** | **0.247** | 0.269 | 0.324 | 0.198 | 0.260 | **0.197** | **0.255** | 0.216 | 0.257 | 0.200 | 0.254 | **0.194** | **0.253** | 0.191 | 0.251 | 0.190 | 0.245 | **0.187** | **0.233** | 11.6 | 2.46 |
| 720 | 0.229 | 0.282 | 0.203 | 0.261 | **0.200** | **0.256** | 0.271 | 0.327 | 0.207 | 0.260 | **0.204** | 0.265 | 0.211 | 0.250 | 0.202 | 0.255 | **0.198** | **0.254** | 0.197 | 0.256 | 0.195 | 0.251 | **0.192** | **0.245** | 11.6 | 1.38 |
| **Wind** | | | | | | | | | | | | | | | | | | | | | | | | | | |
| 96 | 0.186 | 0.216 | 0.175 | 0.213 | **0.170** | **0.207** | 0.197 | 0.230 | 0.177 | 0.214 | **0.174** | **0.212** | 0.172 | 0.218 | 0.169 | 0.208 | **0.168** | **0.202** | 0.170 | 0.211 | 0.169 | 0.208 | **0.167** | 0.210 | 6.01 | 1.80 |
| 192 | 0.204 | 0.229 | 0.191 | 0.224 | **0.187** | **0.219** | 0.218 | 0.245 | **0.187** | 0.226 | 0.193 | **0.225** | 0.189 | 0.230 | 0.187 | 0.222 | **0.187** | **0.213** | 0.186 | 0.223 | 0.184 | 0.220 | **0.181** | **0.219** | 6.14 | 1.61 |
| 336 | 0.216 | 0.239 | 0.202 | 0.234 | **0.199** | **0.230** | 0.233 | 0.258 | **0.205** | 0.238 | 0.205 | **0.239** | 0.201 | 0.240 | 0.199 | 0.232 | **0.198** | **0.231** | 0.195 | 0.233 | 0.192 | 0.227 | **0.188** | **0.224** | 5.90 | 1.03 |
| 720 | 0.231 | 0.253 | 0.215 | 0.247 | **0.212** | **0.244** | 0.254 | 0.278 | 0.225 | 0.256 | **0.218** | **0.251** | 0.254 | 0.278 | 0.225 | 0.256 | **0.221** | **0.251** | 0.200 | 0.236 | 0.200 | 0.232 | **0.197** | **0.229** | 7.68 | 1.38 |
| **PeMSD8** | | | | | | | | | | | | | | | | | | | | | | | | | | |
| 96 | 0.445 | 0.316 | 0.410 | 0.305 | **0.397** | **0.300** | 0.562 | 0.421 | 0.449 | 0.336 | **0.431** | **0.328** | 0.373 | 0.290 | 0.368 | **0.286** | **0.366** | 0.288 | 0.393 | 0.304 | 0.386 | 0.297 | **0.385** | **0.293** | 10.6 | 2.09 |
| 192 | 0.519 | 0.354 | 0.471 | 0.337 | **0.466** | **0.334** | 0.611 | 0.443 | **0.502** | 0.364 | 0.505 | 0.370 | 0.409 | 0.312 | 0.399 | 0.303 | **0.397** | **0.302** | 0.440 | 0.333 | 0.429 | 0.324 | **0.420** | **0.317** | 9.28 | 0.66 |
| 336 | 0.562 | 0.366 | 0.511 | 0.353 | **0.506** | **0.344** | 0.648 | 0.462 | 0.532 | 0.379 | **0.530** | **0.375** | 0.439 | 0.318 | 0.430 | 0.310 | **0.422** | **0.305** | 0.468 | 0.350 | 0.441 | 0.333 | **0.437** | **0.330** | 10.6 | 1.54 |
| 720 | 0.653 | 0.403 | 0.563 | 0.378 | **0.542** | **0.369** | 0.748 | 0.519 | 0.597 | 0.414 | **0.595** | 0.424 | 0.488 | 0.356 | 0.468 | 0.338 | **0.451** | **0.322** | 0.511 | 0.379 | 0.484 | 0.342 | **0.481** | **0.338** | 14.4 | 2.18 |

recorded at 5- or 10-minute intervals, requiring more sophisticated dependency modeling. For instance, although both Traffic and PeMSD8 datasets consists of traffic recordings from a large number of sensors (862 and 510 sensors, respectively), the PeMSD8 dataset exhibits much greater fluctuation and recording errors, making it more difficult for CI models to accurately predict future states by relying solely on temporal dependencies. For channel-dependent (CD) methods, such as Crossformer and MTGNN, CALAS makes an average improvements of 5.34% across six datasets. Notably, Electricity and Traffic datasets show the largest improvements, with gains of up to 18.46%. This is due to the presence of multiple short-term periodicities in the data, which require an understanding of frequency or propagation delay. By injecting an awareness for propagation delay, CALAS significantly enhances the modeling qualities on highly periodic datasets.

**Improvements over plug-in method** Compared to the recently developed plug-in method, LIFT, CALAS achieves an additional improvement of 2.21 percentage points. This improvement is even more pronounced in datasets containing signals from a variety of sensors (e.g., Weather) and in long-term forecasting tasks. We hypothesize that this performance gain is caused by several factors: **F1**: the presence of short-term distribution shifts in channel dependency, **F2**: the limited diversity of statistically extracted 'cause' signals, and **F3**: limitation of LIFT's layer-specific design.

Across all datasets, we observe that CALAS tends to outperform LIFT, particularly in long-term forecasting tasks. We will showcase three cases: Weather, Electricity, and Traffic datasets. Weather dataset consists of signals exhibiting diverse patterns caused by various sources. Moreover, their causality can shift in the short-term period–for example, humidity and temperatures indicate no causality during certain periods but exhibit strong negative causality during the onset of the rainy season. These short-term changes, even between the input and output signals, degrade LIFT's performance. This degradation experimentally proves **F1**: distribution shifts in channel dependency, and **F3**: LIFT's reliance on input to generate output channel dependencies, highlighting the need of layer-agnostic usage of the causality modeling as in CALAS.

Electricity and Traffic datasets are recorded at hourly intervals and have strong daily periodicity, generating multiple short-term periodic patterns. As a result, LIFT, particularly through cross-correlation, tends to focus on capturing the short-term periodicities while overlooking trend changes. Additionally, its top-$k$ selection process with redundant patterns creates an information bottleneck, limiting access to diverse information (**F2**). In contrast, CALAS learns the causal relationships in an end-to-end manner, allowing the model to avoid such bottlenecks and ultimately produce richer forecasting results. Furthermore, we have found that CALAS can support statistics-based model, LightMTS (Zhao & Shen, 2024), by modeling input causality, improving additional 1.2 percentage points over CALAS, 3.41 percentage points with respect to original LightMTS, as Figure 4c.

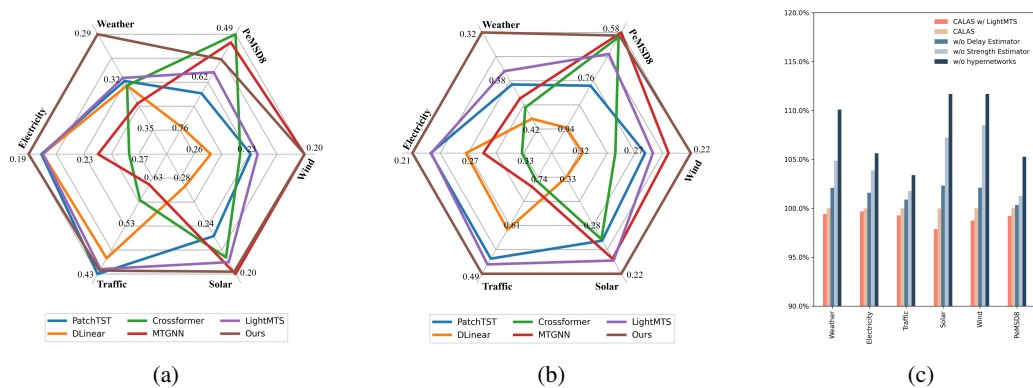

Figure 4: Performance of Linear model-based CALAS (Ours, Brown line) and five baseline models on (a) all variates and (b) top-10% variates with strong causal relationship (i.e., highly affected from the other variates). (c) Performance of CALAS with LightMTS and ablation study on CALAS with hypernetworks, introducing partially (or fully) static causality modeling.

**CALAS with Linear Backbone**   To showcase the importance of dynamic causality modeling, we conduct additional experiments with simple Linear backbone with RevIN (Kim et al., 2022). As shown in Figure 4a, CALAS with one linear projection still shows comparable performance to state-of-the-art models. We also indicate that if certain variates have strong causal relationship, which is extracted with CALAS, our model outperform the existing models by utilizing strong causal information and propagation delay. For example, PeMSD8, Solar, and Wind datasets require more sophisticate modeling, degrading the linear models. In contrast, CALAS gains comparable performance with complicate CD modeling methods and even outperform them for the variates with strong causal relationship, as illustrated in Figure 4b.

## 5.3 Ablation Study

In Figure 4c, we provide the ablation study results with our hypernetworks, reporting relative MSE amplification with respect to CALAS. In this study, we compared CALAS with three variants – 1) without delay estimator, 2) without strength estimator, and 3) without both estimators. We utilize the one layer linear function as backbone model and the performance is reported on average over the prediction length 96, 192, 336, and 720. By ablating hypernetworks, CALAS lost its ability to dynamically modeling causality, as a results, degrading the forecasting quality. We can depict that loosing dynamic causal strength modeling (i.e., w/o strength estimator) induces inability to distribution shifts on CD, showing inferior performance. The delay estimator, in contrast, often shows less performance drop than strength estimator, since CALAS still can dynamically estimate causality. Furthermore, the ablation results of CALAS w/o hypernetworks imply the need of transition from static causality modeling to dynamic causality modeling.

## 6 Conclusion

In this work, we emphasize the importance of **delay learning** in channel dependency modeling and propose new perspective for understanding channel dependencies in MTS forecasting. We introduce a novel end-to-end learning method, CALAS, which learns underlying lead-lag relationship in MTS while considering both causal strength and propagation delay. By integrating Gaussian kernel-based 1D convolution with two hypernetworks, CALAS reformulate undirected channel dependency modeling into directed, delay-aware causality modeling. We experimentally demonstrate the superiority of CALAS over state-of-the-art MTS forecasting models and statistical causality modeling methods. We believe that delay learning and causality-aware modeling are promising research directions for future developments of MTS forecasting models. As a next step, we plan to investigate the better model architecture for delay learning, including multi-periodicity decomposition by adjusting stride or introducing the uncertainty concepts into the CD modeling.

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

Table 1: Performance on TSCD task on CausalTime datasets. We utilize the performance of baseline TSCD algorithms reported in CausalTime paper (Cheng et al., 2024). **Bold text** means the best model and underlined text indicates second-best model

| Methods | AUROC | | | AUPRC | | |
|---|---|---|---|---|---|---|
| | AQI | Traffic | Medical | AQI | Traffic | Medical |
| GC | $0.4538 \pm 0.0377$ | $0.4191 \pm 0.0310$ | $0.5737 \pm 0.0338$ | $0.6347 \pm 0.0158$ | $0.2789 \pm 0.0018$ | $0.4213 \pm 0.0281$ |
| SVAR | $0.6225 \pm 0.0406$ | $\underline{0.6329} \pm 0.0047$ | $0.7130 \pm 0.0188$ | $0.7903 \pm 0.0175$ | $0.5845 \pm 0.0021$ | $0.6774 \pm 0.0358$ |
| N.NTS | $0.5729 \pm 0.0229$ | $\mathbf{0.6329} \pm \mathbf{0.0335}$ | $0.5019 \pm 0.0682$ | $0.7100 \pm 0.0228$ | $0.5770 \pm 0.0542$ | $0.4567 \pm 0.0162$ |
| PCMCI | $0.5272 \pm 0.0744$ | $0.5422 \pm 0.0737$ | $0.6991 \pm 0.0111$ | $0.6734 \pm 0.0372$ | $0.3474 \pm 0.0581$ | $0.5082 \pm 0.0177$ |
| Rhino | $0.6700 \pm 0.0983$ | $0.6274 \pm 0.0185$ | $0.6520 \pm 0.0212$ | $0.7593 \pm 0.0755$ | $0.3772 \pm 0.0093$ | $0.4897 \pm 0.0321$ |
| CUTS | $0.6013 \pm 0.0038$ | $0.6238 \pm 0.0179$ | $0.3739 \pm 0.0297$ | $0.5096 \pm 0.0362$ | $0.1525 \pm 0.0226$ | $0.1537 \pm 0.0039$ |
| CUTS+ | $\mathbf{0.8928} \pm \mathbf{0.0213}$ | $0.6175 \pm 0.0752$ | $\mathbf{0.8202} \pm \mathbf{0.0173}$ | $\underline{0.7983} \pm 0.0875$ | $\underline{0.6367} \pm 0.1197$ | $0.5481 \pm 0.1349$ |
| NGC | $0.7172 \pm 0.0076$ | $0.6032 \pm 0.0056$ | $0.5744 \pm 0.0096$ | $0.7177 \pm 0.0069$ | $0.3583 \pm 0.0495$ | $0.4637 \pm 0.0121$ |
| NGM | $0.6728 \pm 0.0164$ | $0.4660 \pm 0.0144$ | $0.5551 \pm 0.0154$ | $0.4786 \pm 0.0196$ | $0.2826 \pm 0.0098$ | $0.4697 \pm 0.0166$ |
| LCCM | $0.8565 \pm 0.0653$ | $0.5545 \pm 0.0254$ | $0.8013 \pm 0.0218$ | $\mathbf{0.9260} \pm \mathbf{0.0246}$ | $0.5907 \pm 0.0475$ | $\mathbf{0.7554} \pm \mathbf{0.0235}$ |
| eSRU | $0.8229 \pm 0.0317$ | $0.5987 \pm 0.0192$ | $0.7559 \pm 0.0365$ | $0.7223 \pm 0.0317$ | $0.4886 \pm 0.0338$ | $0.7352 \pm 0.0600$ |
| SCGL | $0.4915 \pm 0.0476$ | $0.5927 \pm 0.0553$ | $0.5019 \pm 0.0224$ | $0.3584 \pm 0.0281$ | $0.4544 \pm 0.0315$ | $0.4833 \pm 0.0185$ |
| TCDF | $0.4148 \pm 0.0207$ | $0.5029 \pm 0.0041$ | $0.6329 \pm 0.0384$ | $0.6527 \pm 0.0087$ | $0.3637 \pm 0.0048$ | $0.5544 \pm 0.0313$ |
| CALAS | $\underline{0.8772} \pm 0.0287$ | $0.6312 \pm 0.0461$ | $\underline{0.8124} \pm 0.0125$ | $0.6788 \pm 0.0512$ | $\mathbf{0.6701} \pm \mathbf{0.0980}$ | $0.7412 \pm 0.0518$ |

# A  ADDITIONAL EXPERIMENTAL RESULTS

To prove that CALAS actually finds the ground truth causality, we conduct experiments with three real-world datasets in CausalTime benchmark (Cheng et al., 2024) and one well-known Synthetic dataset for causal discovery (Suiz A. Baccalá, 2001). We quantitatively and qualitatively showcases CALAS's superiority on causal discovery with Air-quality (AQI), Traffic, and Medical datasets, experimentally proving that CALAS can actually model the causal relationship. We compared CALAS with various baselines including: Granger causality (GC) (Granger, 1969), neural Granger causality (NGC) (Tank et al., 2022),economy-SRU (eSRU) (Khanna & Tan, 2020), scalable causal graph learning (SCGL) (Xu et al., 2019), temporal causal discovery framework (TCDF) (Nauta et al., 2019), CUTS (Cheng et al., 2023b), CUTS+ (Cheng et al., 2023a), PCMCI (Runge et al., 2019), SVAR, NTS-NOTEARS (shown as N.NTS) (Sun et al., 2023), Rhino (Gong et al., 2023), latent convergent cross mapping (LCCM) (Brouwer et al., 2021), and neural graphical model (NGM) (Bellot et al., 2022). For the synthetic dataset, we only provide visual comparison among Granger causality test, LIFT (i.e., cross-correlation), and CALAS. In the causal discovery experiments, we stick to the our MTS forecasting setting with input length 336 but with output length 1. As a backbone, we utilize one layer linear model.

## A.1  QUANTITATIVE EVALUATION ON CAUSAL DISCOVERY

Table 1 indicates the experimental results for the time series causal discovery (TSCD) task with three real-world datasets. Even though CALAS focuses on the dynamic causality discovery, it exhibits competitive results across all three datasets in traditional TSCD task, achieving the best performance in AUPRC for the Traffic dataset, and competitive performance to the state-of-the-art models, such as CUTS+, in AUROC and AUPRC across Medical and AQI datasets. It experimentally proves that CALAS successfully models the causal relationships during its optimization. Furthermore, CALAS is the one of two algorithms that simultaneously models the propagation delay and causal strength, however, the other one (i.e., TCDF) indicates its limitation to properly model both characteristics. Lastly, despite CALAS has only one the simple hyperparameter, the maximum delay $k$, for the causal modeling, it outperforms the algorithms requiring sophisticated, data-specific hyperparameter settings. It further reduces the difficulty to introduce the algorithm to unseen datasets. However, as we can depict in Figure 1, CALAS refers to improper cause signals, which could be improved by introducing contrastive learning methods or regularization term in optimization.

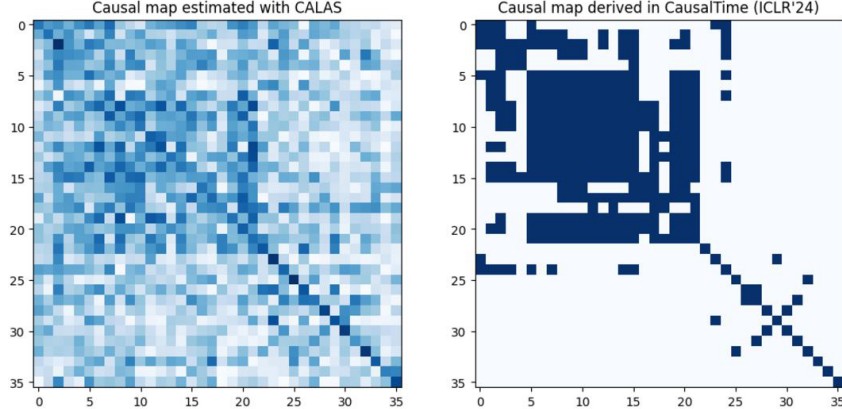

Figure 1: Causality map estimated via CALAS (left) and ground truth causal graph (right).

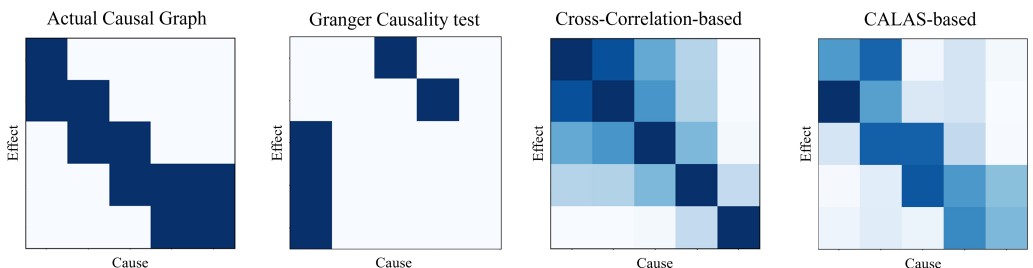

Figure 2: Visualization of actual causal graph and those calculated with Granger causality test, cross-correlation, and CALAS without any hyperparameter tuning or additional techniques like weight decay or $L_2$ normalization.

## A.2 QUALITATIVE EVALUATION ON CAUSAL DISCOVERY

To prove that CALAS actually finds the ground truth causality, we conduct experiments with a well-known Synthetic dataset for causal discovery (Suiz A. Baccalá, 2001). We have compared CALAS with Granger Causality test and LIFT (i.e., cross-correlation). We have excluded other CD modeling methods in MTS forecasting, because 1) they are unable to model propagation delay, which is unsuitable for causal discovery, as we depicted in main paper. Furthermore, to compare the performance in nature, that means, end-to-end manner without *any hyperparameter tuning*, we have excluded deep learning-based causal discovery methods, such as TCDF (Nauta et al., 2019), CUTS (Cheng et al., 2023b), cLSTM (Tank et al., 2022), or other methods. Please note that causal discovery models require sophisticate hyperparameter tuning to obtain a proper causality graph (Nauta et al., 2019; Li et al., 2023).

**Main results** Figure 2 indicates the actual causal graph and those calculated with each methods. It indicates that CALAS can successfully approximate actual causality graph without any hyperparameter tuning related to optimization or model parameter. It indicates both validity and importance of learning propagation delay, which has not yet been investigated in both causal discovery and MTS forecasting. It also provides another lesson-learned to the causal discovery domain that we do not actually need to data-specifically and manually conduct hyperparameter search and only need to optimize both propagation delay and causal strength. Please note that this phenomenon is also reported as one of the challenges in CD modeling for MTS forecasting–model tend to encounter overfitting issue without delay estimation or dynamic CD modeling (Han et al., 2023).

Table 2: Performance comparison in terms of forecasting errors. The **bold text** means the best results and underlined text means the second best results.

| Method | | TimesNet | | CALAS +Linear | | DLinear | |
|---|---|---|---|---|---|---|---|
| | | MSE | MAE | MSE | MAE | MSE | MAE |
| Weather | 12.5% | **0.025** | **0.045** | **0.025** | **0.045** | 0.039 | 0.101 |
| | 25.0% | **0.029** | 0.052 | 0.030 | **0.051** | 0.048 | 0.111 |
| | 37.5% | **0.031** | **0.057** | 0.033 | 0.065 | 0.057 | 0.121 |
| | 50.0% | **0.034** | **0.062** | 0.040 | 0.072 | 0.066 | 0.134 |
| Electricity | 12.5% | 0.085 | 0.202 | **0.063** | **0.170** | 0.092 | 0.214 |
| | 25.0% | 0.089 | 0.206 | **0.077** | **0.190** | 0.118 | 0.247 |
| | 37.5% | 0.094 | 0.213 | **0.093** | **0.206** | 0.144 | 0.276 |
| | 50.0% | **0.100** | **0.221** | 0.106 | 0.230 | 0.175 | 0.284 |

### A.2.1 DATA GENERATION

For the training, we utilize Synthetic dataset generated with following equations:

$$
\begin{cases}
x_1(n) = 0.95\sqrt{2}x_1(n-1) - 0.9025x_1(n-2) + w_1(n) \\
x_2(n) = -0.5x_1(n-1) + w_2(n) \\
x_3(n) = 0.4x_2(n-2) + w_3(n) \\
x_4(n) = -0.5x_3(n-1) + 0.25\sqrt{2}x_4(n-1) + 0.25\sqrt{2}x_5(n-1) + w_4(n) \\
x_5(n) = -0.25\sqrt{2}x_4(n-1) + 0.25\sqrt{2}x_5(n-1) + w_5(n)
\end{cases}
, \qquad (5)
$$

where $n$ is $n$-th time steps, $x_i$ means $i$-th variate, and $w_i(n)$ means zero-mean uncorrelated white processes with identical variances.

### A.3 EXPERIMENTAL RESULTS FOR IMPUTATION TASK

## B DISCUSSION

**Discussion on other CNN- or RNN-based methods** The emergence of Mamba (Gu & Dao, 2024) played a significant role in shifting researchers' focus from Transformer models to State Space Models (SSMs). One main flow in the SSM research is the extension of 1D Mamba to the multidimensional state spate models. In the MTS forecasting, there are RNN-based (Tank et al., 2022; Jia et al., 2023; Behrouz et al., 2024), CNN-based (Wu et al., 2023), and SSM-based approaches (Zeng et al., 2024; Hu et al., 2024). In the following paragraphs, we will discuss how they can achieve inter-variate dependency modeling and difference between CALAS and them.

*RNN-based approaches*, for example, WITRAN (Jia et al., 2023) or cRNN (Tank et al., 2022), need to propose additional channel dependency module to model the inter-variates relationships. In case of WITRAN, similar to the TimesNet (Wu et al., 2023), it folds the input 1D time series into 2D time series, enabling intra- and inter-periodicity modeling. However, it less focuses on the potential causal effects from the other time series. cRNN (Tank et al., 2022), one of the causal discovery algorithms, induces the inter-variates relationships via weights of trained RNN (i.e., projection layer of RNNs), however, it only captures the gradual changes, especially with the one-step time lag.

For the *CNN- and SSM-based approaches*, such as C-Mamba (Zeng et al., 2024) or TimeSSM (Hu et al., 2024), the main difference between CALAS and these approaches are how they dealt with receptive fields and introduces inductive biases into model. C-Mamba and TimeSSM considers the convolutional- or SSM-based state space as the range of information fusion, not introducing inductive biases into. For example, in case of 1D $\mathbb{R}^{N_i \times N_o \times k}$ CNN kernel, where $N_i, N_o, k$ are input and output channel and kernel size, respectively, aforementioned model optimize both causal strength and propagation delay into one kernel. In such design, fused with causal strength, propagation delay will be independent weights to each other. However, for the inter-variate relationship, there are only one unique propagation delay, which should be modeled with probability function among $k$ possible delays. CALAS disentangles the propagation delay and approximate them with Gaussian probability

kernel, it introduces additional inductive bias–given two cause signal $X$ and effect signal $Y$, there exists unique discrete delay $d_{X,Y}$ such that the time gap between change of $X$ and its actual influence to $Y$.

**Transformer-based methods vs. CALAS**  Transformer-based methods, including iTransformer, mix the multivariate information regardless of propagation delay. This design may introduce misaligned or outdated information from lagged time series, resulting to degrade of model performance. To properly align the variates, previous models should borrow the modeling capacity from temporal dependency modeling components, which lowers the temporal dependency modeling quality. By facilitating proposed convolution, CALAS simultaneously conduct such alignment and CD modeling. Though LIFT (Zhao & Shen, 2024) achieves lead-lag relation modeling with cross-correlation, it requires additional computation and relies on the statistical methods that often be suboptimal.

**Layer-agnostic causality modeling is important in MTS forecasting**  Distribution shifts of the statistical features are well-studied problem in MTS forecasting (Kim et al., 2022; Zeng et al., 2023; Liu et al., 2022). However, distribution shifts of channel dependency is not yet investigated deeply. Here, we derive a discussion for distribution shifts in short period, as we depicted in Figure 1. By introducing shared CALAS across multiple layers except the input-dependent parts, our model reduces the misalignment or over-reliance of previously generated causal maps when short term distribution shifts occur.

**Generalization for multi-periodicity modeling**  Since CALAS stems from convolution mechanism, we can achieve the multi-periodicity decomposition by adjusting the stride. However, addressing this question is beyond the scope of this work, so we leave it for future exploration.

