# OpenReview forum: "Learning by Causality to Improve Channel Dependency Modeling in Multivariate Time Series Forecasting"
_ICLR.cc/2025/Conference — Submitted to ICLR 2025_

### Official Review · Reviewer_x5dP · 2024-10-31

**Soundness:** 1
**Presentation:** 2
**Contribution:** 1
**Rating:** 3
**Confidence:** 3

**Summary:**

This paper introduces CALAS, a new attention-style mechanism which is able to achieve improved performance across several datasets by combining DLCS (dilated convolutions with learnable spacing) alongside two hypernetwork learners to learn the lead-lag relationships in the time series dataset. Empirical performance is compared against the major prior work, LIFT, and competitive performance is achieved.

**Strengths:**

The paper targets an important problem of understanding channel dependencies and specifically follows the line of research highlighted by the LIFT paper.  This approach seems more structured than the existing LIFT paper by the use of the specific functional form in Equation (1); however, the nuances of such an assumption and its relation with existing work are not sufficiently explored (see below).

**Weaknesses:**

The authors use the language of "uncovering causality" and yet gravely misunderstand the existing literature in both Granger causality and causal discovery.  In this sense, the research is about the channel dependencies using the same flavor of cross-temporal dependencies as the original definition of Granger causality.  In spite of this, no discussion about this approach is given. Moreover, the language of "true causality" is used throughout the paper, despite it being known for decades that this is not true causality.  It should be noted that this alone can be considered as sufficient grounds for rejection.

Other less serious concerns include:
- performance improvements over LIFT are generally extremely marginal and without error bars
- no analysis is given on the final learned models and the 'causal' structure which they uncover
- no theoretical analysis is given regarding the causal discovery of the proposed approach

**Questions:**

Can you clarify the graphical assumptions which are being made onto the multivariate time series to allow easier comparison with existing works in causality for time series?

Under what assumptions can you guarantee the exact recover and/ or identifiability of the underlying causal structure which you are aiming to learn?

Can you clarify how existing work in causality and in Granger causality compare to your proposed work?

Can you identify what you see as the key innovation of your work compared to the existing LIFT work?

---

> ### Author Response · Authors · 2024-12-02
> **Response to Reivewer x5dP (Part 1)**
>
> We appreciate to your constructive comments. Based on your comments, we can improve our manuscripts significantly. We want to provide the detailed explanation corresponding to your concerns and questions.
>
> ## W1. The authors use the language of "uncovering causality" and yet gravely misunderstand the existing literature in both Granger causality and causal discovery. In this sense, the research is about the channel dependencies using the same flavor of cross-temporal dependencies as the original definition of Granger causality. In spite of this, no discussion about this approach is given.
>
>
> > W1-A1.  Thank you for your comments! Our methods indeed stem from the Granger Causality. We will revisit the Granger causality and clarify the “causality” in our paper. Yet, please note that the “causality” in our paper is not exactly the same as the one in the causal discovery – we will highlight it in Section 4.
>
> > Furthermore, we want to emphasize that “The preceding information (i.e., lead-lag relationship) is crucial for MTS and should be considered in the model” is a common and experimentally proven quote in MTS forecasting. For example, in traffic forecasting, DCRNN [1], MTGNN [2], and the other traffic domain articles [3] highlight the importance of preceding information, and the recent MTs literature further supports it [4-7].
>
> ## W2. Moreover, the language of "true causality" is used throughout the paper, despite it being known for decades that this is not true causality. It should be noted that this alone can be considered as sufficient grounds for rejection.
>
> > W2-A1. We would like to note that we didn’t use any term for “true causality” in the paper. It would be beneficial if you can point out the specific part that makes you confused. However, the definition of causality in this paper is closely related to the causality in causal discovery and Granger causes, so we clarified them in Section 4. Furthermore, in the time series, such as traffic, there are many articles that depict the importance of the lead-lag relationship and propagation delay, which is necessarily discussed in the channel-dependency modeling of the time series [1-7].
>
>
> ## W3. marginal performance improvement, no analysis on causal structure, no theoretical analysis for causal discovery
>
> > W3-A1. Thank you for your comment. We clarify them in our paper. We provide detailed answers below, each of which is clarified in the revised version as well.
>
> > (About performance improvements) Since LIFT and CALAS are extremely stable in MTS forecasting tasks (less than 0.001 deviations), the improvement of 2.21 percent points is not marginal and statistically significant (Section 5.2)
> > (About causal discovery and corresponding theoretical and experimental analysis) We briefly describe our concepts based on the Granger causality test in Section 4. Moreover, we experimentally prove the causal structures are captured via CALAS by conducting causal discovery tasks in Appendix A.
>
>
> [1] Li et al., “Diffusion Convolutional Recurrent Neural Network: Data-Driven Traffic Forecasting,” ICLR 2018
>
> [2] Wu et al., “Connecting the dots: Multivariate time series forecasting with graph neural networks,” KDD 2020
>
> [3] Jin et al., “A visual analytics system for improving attention-based traffic forecasting models,” VIS 2023
>
> [4] Lifan Zhao and Yanyan Shen, “Rethinking channel dependence for multivariate time series forecasting: Learning from leading indicators,” ICLR 2024
>
> [5] Zeng et al., “C-Mamba: Channel Correlation Enhanced State Space Models for Multivariate Time Series Forecasting”, [J]. arXiv preprint arXiv:2406.05316, 2024.
>
> [6] Luo Donghao and Wang Xue, “Moderntcn: A modern pure convolution structure for general time series analysis”, ICLR 2024.
>
> [7] Hu et al., Time-SSM: Simplifying and Unifying State Space Models for Time Series Forecasting”, [J]. arXiv preprint arXiv:2405.16312, 2024.

---

> > ### Author Response · Authors · 2024-12-02
> > **Response to Reivewer x5dP (Part 2)**
> >
> > ## Q1. Under what assumptions can you guarantee the exact recover and/ or identifiability of the underlying causal structure which you are aiming to learn?
> >
> > > Q1-A1. The concepts and learning theory are closely related to the Granger causality – we have explained them in Section 4. Furthermore, we experimentally prove the identified causal structure in Appendices A.2 and A.3.
> > > More specifically, we built the definition of causality with three assumptions: 1) It should follow the Markovian condition [1, 2], 2) causal faithfulness [1], and 3) should have no instantaneous effects [3]. Since real-world datasets are mostly constructed with only the signals from the sampled set of sensors, we have not assumed causal sufficiency and stationarity (in Pearl 2009, stability).
> >
> > ## Q2. Can you clarify how existing work in causality and in Granger causality compare to your proposed work?
> >
> > > Q2-A1. CALAS and causal discovery methods have different assumptions and definitions for causality. In this paper, we develop causality based on commonly utilized assumptions except for sufficiency and stationarity. Also, the definition of “causality” in this paper stems from the Granger causality with a more detailed and specialized module for lag estimation. Instead of estimating the coefficient for each time lag, we approximate it with the probability function (i.e., Gaussian kernel), enabling better causality modeling. We clarified it in Section 4.1.
> >
> > ## Q3. Can you identify what you see as the key innovation of your work compared to the existing LIFT work?
> >
> > > Q3-A1. First of all, LIFT relies on the input-output relationship, which reduces the flexibility of the methods and modeling diversity. For example, in the highway traffic data, LIFT mostly focuses on multiple similar patterns, which are highly influenced by small white noises. Furthermore, it is only capable of the signals that could be properly modeled with cross-correlation. In contrast, CALAS is learning-based, which avoids over-relying on multiple similar patterns, and is capable of learning most of the signals including pulse signal.
> >
> > > Lastly and most importantly, CALAS can estimate the propagation delay by learning, which is hardly discussed in LIFT. Those differences are clarified in Section 5, “Improvements over plug-in methods,” which directly compares CALAS and LIFT experimentally.
> >
> >
> > [1] Judea Pearl, “Causality: Models, Reasoning, and Inference,” Cambridge University Press, New York. 2nd Edition, 2009
> >
> > [2] Judea Pearl, “Causal Inference in Statistics: An overview,” Statistics Surveys, 2009
> >
> > [3] Assaad et al., “Survey and evaluation of causal discovery methods for time series,” Journal of Artificial Intelligence Research, 2022

---

### Official Review · Reviewer_GX4s · 2024-11-03

**Soundness:** 2
**Presentation:** 2
**Contribution:** 2
**Rating:** 3
**Confidence:** 3

**Summary:**

This paper reanalyzes the task of multivariate time series forecasting from the perspectives of causal strength and propagation delay. This is highly meaningful for improving model performance and interpretability. Moreover, this method has the advantage of adaptive training compared to previous statistical approaches. However, the articulation of this article is not very clear, and due to the absence of open-source code, reproducing the results is challenging. Additionally, the experimental section is not comprehensive enough.

**Strengths:**

The topic in this paper is meaningful. Modeling multivariate time series causality from the perspective of delays is notable. Parameterizing this process with a Gaussian kernel enables the model to adaptively explore the delayed causal relationships among multiple variables.

**Weaknesses:**

* **W1**: I believe the primary issue with this article lies in its lack of clarity in expression and the substantial absence of details within the text (for more information, refer to the "Questions" section). Moreover, the absence of corresponding code in the supplementary materials makes it challenging to complete the model framework and reproduce the results.

* **W2**: The experimental evaluation in this paper is weak, as it only encompasses standard prediction task results and ablation studies, neglecting the interpretability analysis emphasized in the Introduction. I suggest that the authors include more case studies and explore the effectiveness of the proposed modules in tasks such as classification, anomaly detection, and others.

* **W3**: Currently, there are numerous models that incorporate specialized modules [1,2] to model variable causality on top of modeling time dependencies with a CI strategy. In particular, conducting Fourier transforms on the variable dimensions and utilizing linear layers for learning can efficiently and lightly model variable relationships [3]. The authors should comprehensively compare the advantages and disadvantages of these approaches, including their efficiency.

* **W4**: Some minor details to consider include avoiding the use of "you" in expressions such as in line 321, and ensuring that each line does not consist of only a few words.


[1 ]Zeng C, Liu Z, Zheng G, et al. C-Mamba: Channel Correlation Enhanced State Space Models for Multivariate Time Series Forecasting[J]. arXiv preprint arXiv:2406.05316, 2024.

[2] Luo D, Wang X. Moderntcn: A modern pure convolution structure for general time series analysis[C]//The Twelfth International Conference on Learning Representations. 2024.

[3] Hu J, Lan D, Zhou Z, et al. Time-SSM: Simplifying and Unifying State Space Models for Time Series Forecasting[J]. arXiv preprint arXiv:2405.16312, 2024.

**Questions:**

* **Q1**: The paper has no corresponding content regarding how to penalize $\sigma$, as mentioned in line 337.

* **Q2**: The statement from lines 347 to 366 is quite ambiguous. Phrases like "the causal strength is shared for k possible delays" lack clarity. Terms such as the embedding vector M, h, and $N_o$ are undefined. I suggest that the authors rephrase this section for better understanding and coherence.

* **Q3**: Section 4.2 does not clearly explain how to handle multivariate time series data after obtaining the causal graph. Additionally, it only covers causal weight generation, but it does not detail how the other two parameters, $\sigma$ and d, of the Gaussian kernel are generated. These missing aspects need to be addressed in the paper.

---

> ### Author Response · Authors · 2024-12-02
> **Response to Reviewer GX4s**
>
> Thank you for your detailed review! We have updated our manuscripts accordingly. We want to provide detailed responses for your concerns and questions below:
>
> ## W1. Lack of clarity in expression, Absence of corresponding code
>
> > W1-A1. We clarified the expressions (See the detailed answers below). The corresponding codes will be published to the public after the acceptance
>
>
> ## W2. Experimental results in this paper are shallow. The interpretability analysis and case studies are required.
>
> > W2-A1. Since the MTS forecasting datasets have no ground truth causal graphs, we have conducted additional experiments with causal discovery benchmarks published in CausalTime (ICLR’24). We have also conducted experiments with synthetic datasets and two widespread statistical measures, the Granger causality test and Correlation-based methods (i.e., LIFT and PDC [2, 3]).
>
> > We updated the results on the imputation task on Electricity and Weather datasets in Table 2, Appendix A.3. We referred to the setting of TimesNet paper results. We are updating the remaining tasks and will report them in the camera ready (if allowed).
>
>
> ## W3. The author should comprehensively compare the advantage and disadvantages of previous approaches.
>
> > W3-A1. Thank you for your valuable comment! The main difference between those methods and our methods is the view of receptive fields; CALAS considers receptive fields as a possible time lag and conducts Gaussian kernel-based lag modeling—which is a continuous approximation of the pulse function focused on time-lag information modeling. This design introduces inductive bias of discrete time lag of cause-effect relationship into the convolutional mechanism. In contrast, previous models consider receptive fields as the range of information fusion w/o any constraint. It provides an extra modeling range for the model but also introduces difficulty to modeling time lag and causal information of time series. We have updated the references and added a discussion for comparison between SSM-based methods and CALAS in Appendix B.
>
> ## W4. Minor details
>
> > W4-A1. We have revised it accordingly.
>
>
>
> ## Q1. The paper has no corresponding content regarding how to penalize $\sigma$
>
> > Q1-A1. Starting with a large σ value, we penalized it until it reached its minimum value of 0.5, as Hammouamri et al. (2024). Specifically, we introduced the upper bound for σ, which becomes smaller during training.
>
>
> ## Q2. The statement in Section 4.2, hypernetworks and weight generation process is ambiguous and some terms are not defined.
>
> > Q2-A1. We clarify the ambiguous and undefined terms. Also, we referred to the previously introduced equations for clarity.
>
> ## Q3. Section 4.2 does not clearly explain how to handle MTS data after obtaining the causal graph.
>
> > Q3-A1. After we generate the causal strength and propagation delay, the propagation process is followed, which we depicted in Section 4.1. Also, the generation process σ and d are mostly the same as the process of the causal strength. We add a more detailed explanation in Section 4.
>
>
> [1] Cheng et al., “CausalTime: Realistically generated time-series for benchmarking of causal discovery,” ICLR 2024
>
> [2] Luiz A. Baccalá and Koichi Sameshima, “Partial directed coherence: a new concept in neural structure determination,” Biological Cybernetics, 2001
>
> [3] Lifan Zhao and Yanyan Shen, “Rethinking channel dependence for multivariate time series forecasting: Learning from leading indicators,” ICLR 2024

---

### Official Review · Reviewer_ivnz · 2024-11-03

**Soundness:** 3
**Presentation:** 3
**Contribution:** 3
**Rating:** 8
**Confidence:** 4

**Summary:**

This paper has presented CALAS, an end-to-end method for multivariate timeseries forecasting that can learn causality among variates without any statistical measures or prior knowledge. CALAS decomposes causal effect into causal strength and propagation delay, this results in better capturing the time delay and better interpretability, without adding any additional complexity to the model. The experimental results show that CALAS can provide more improvement than non-learning methods like LIFT.

**Strengths:**

- One of the important factors in measuring causality is the time required to wait to see the effect. While it has been extensively discussed, I believe estimating the delay as a part of the model is understudied. The idea of delay estimation is indeed important and interesting.
- In addition to the good performance of CALAS in improving the chosen baselines, it is more interpretable than existing methods as it helps to understand causal strength and propagation delay separately.
- The paper is overall well-motivated and easy to follow, though the presentation requires improvements.
An ablation study has been conducted and has shown the importance of contributions.

**Weaknesses:**

- The presentation in the paper can be greatly improved. While the overall writing is good and contributions are well-motivated, there are several repeated sections that can be removed and used for additional experimental results or making figures bigger. For example, lines 88 and 103 are the same. The contributions are already discussed in lines 145-160, so there is no need to repeat them in 209-215. I also found Section 6 (Discussion) unnecessary and redundant as it has mostly been discussed in the paper.
- It seems that the authors have overlooked a part of the literature that uses causal RNN-style (or convolution-style) models across variates [1, 2, 3]. Contrary to transformer-based models, these methods can provide us with directional causality.
- The main focus of the paper and experimental results are relatively narrow. Most recent studies on multivariate time series are more general methods [1] and achieve good performance across diverse tasks like classification, imputation, and anomaly detection. I do not see a limitation for CALAS to be applied for other tasks and it would be great if the authors could enhance the experimental results.

---
[1] TimesNet: Temporal 2D-Variation Modeling for General Time Series Analysis. Wu et al., ICLR 2023.
[2] WITRAN: Water-wave Information Transmission and Recurrent Acceleration Network for Long-range Time Series Forecasting. Jia et al., NeurIPS 2023.
[3] Chimera: Effectively Modeling Multivariate Time Series with 2-Dimensional State Space Models. Behrouz et al., NeurIPS 2024.

**Questions:**

Please see weaknesses.

---

> ### Comment · Reviewer_ivnz · 2024-11-25
>
> Dear authors,
>
> The end of the discussion period is approaching and it would be great if you could provide your responses for the above points. I am more than willing to increase my score if the above issues can be addressed (specifically additional experimental results).

---

> ### Author Response · Authors · 2024-12-02
> **Response to Reviewer ivnz**
>
> First of all, sorry for the late responses. We spend time to additionally conduct the experiments for causal discovery and imputation tasks, and we are glad to solve your concerns and questions. The detailed updates according to your concerns are below:
>
> ## W1. The paper presentation can be greatly improved. Discussion is unnecessary and redundant.
>
> > W1-A1. Thank you for your comment. We revised the paper accordingly. Also, we move the discussion section to Appendix B.
>
>
> ## W2. The author may overlook a part of literature that use causal RNN- or CNN-style models.
>
> > W2-A1. Thank you for pointing this out! We add the references in the comparison and categorization in the introduction and related works. Furthermore, we provide detailed comparison in Appendix B, discussion.
>
> > Specifically, the main difference between those methods and our methods is the view of receptive fields; CALAS considers receptive fields as a possible time lag and conducts Gaussian kernel-based lag modeling—which is continuous approximation of the pulse function focused on time-lag information modeling. This design introduces inductive bias of discrete time lag of cause-effect relationship into the convolutional mechanism. In contrast, previous models consider receptive fields as the range of information fusion w/o any constraint. It provides an extra modeling range for the model, but also introduces difficulty to modeling time lag and causal information of time series. We have updated the references and added a discussion for comparison between CNN-, RNN-, and SSM-based methods and CALAS in Appendix B.
>
>
> ## W3. The experimental results are relatively narrow.
>
> > W3-A1. We updated the results for the imputation task on Electricity and Weather datasets in Table 2, Appendix A.3. We referred to the setting of TimesNet paper results. We are updating the remaining tasks and will report them in the camera ready version (if allowed).

---

> > ### Comment · Reviewer_ivnz · 2024-12-03
> >
> > I thank the authors for their responses to my comments. I appreciate adding additional experimental results as well as discussion about recent studies. I have no further questions. Following the authors response, I have increased my scores.

---

### Official Review · Reviewer_2T1p · 2024-11-04

**Soundness:** 3
**Presentation:** 1
**Contribution:** 2
**Rating:** 5
**Confidence:** 4

**Summary:**

The paper proposes CALAS, an end-to-end module that captures causality with propagation delay among variates to enhance multivariate time series forecasting. Technically, hypernetworks first output the strength and delay matrices for a given multivariate instance. Then Gaussian kernel is used to approximate the undifferentiable propagation delay. Experiments show that the proposed CALAS enhances forecasting accuracy of different backbones and also outperforms previous plug-in method LIFT.

**Strengths:**

1. Explicitly modeling propagation delay captures the lead-lag structure of causal relationships, allowing CALAS to more accurately reflect how signals influence each other over time.

2. CALAS demonstrates efficient improvements over baseline models.

**Weaknesses:**

1. The fundamental problem of this work is what exactly "causality" is. In Judea Pearl's causal inference framework, causality has a rigorous definition through concepts like do-calculus. However, while this paper frequently refers to causality, it lacks a precise, formal definition of the concept.

2. Building on the previous point, the proposed method appears to be a channel mixing with delay effect. There is no evidence that this model truly learns causality as claimed, nor is there a clear distinction from "correlation."

3. The writing needs improvement. As a technical paper, this work uses overly vague language rather than precise formulas. For example, 3.5 pages are devoted to the background, while the methodology is covered in just one page. In the experimental analysis, speculative phrases like "may show no causality" and "is likely due to" are frequently used without evidence.

**Questions:**

1. How are the extracted strength and delay matrices used in forecasting? Are they used to pre-process the input instance, and then the processed instance is input to the backbone?

2. Please describe the ablation study setting in detail. Including: 1)which backbone is used; 2)which prediction length is evaluated; 3)how are the  strength and delay matrices set in the ablated versions.

3. See my other questions in the #Weaknesses section.

---

> ### Author Response · Authors · 2024-12-02
> **Response to Reviewer 2T1p**
>
> Thank you for your valuable feedback! We have updated our manuscripts accordingly and want to response to the weaknesses and questions you suggested.
>
> ## W1. What exactly "Causality" in this paper is? You need more precise, formal definition for this concept.
>
> > W1-A1. Thank you for pointing this out! We additionally described the scope and definition of causality (in our paper) in Section 4.1. In this paper, we build the causality based on commonly utilized assumptions except for sufficiency and stationarity. Also, the definition of “causality” in this paper stems from the Granger causality with a more detailed and specialized module for lag estimation. Instead of estimating the coefficient for each time lag, we approximate it with the probability function (i.e., Gaussian kernel), enabling better causality modeling.
>
> > More specifically, we built the definition of causality with three assumptions: 1) It should follow the Markovian condition [1, 2], 2) causal faithfulness [1], and 3) should have no instantaneous effects [3]. Since real-world datasets are mostly constructed with only the signals from the sampled set of sensors, we have not assumed causal sufficiency and stationarity (in Pearl 2009, stability).
>
> > Upon those assumptions, we hypothesized that MTS datasets have unseen compounder, event, or both, therefore, CALAS aims to estimate the causal graph of the current MTS snapshot (i.e., $X_t$).
>
> ## W2. There is no evidence that this model truly learns causality as claimed, nor is there a clear distinction from correlation.
>
> > W2-A1. We have additionally conducted experiments with causal discovery datasets and proved that CALAS could capture causality in Appendix A. Furthermore, we provide an explanation for the distinction between correlation, CNN—or RNN-based methods, and CALAS in Appendix B.
>
>
> ## W3. The writing needs improvements
>
> > W3-A1. We fully revised the vague terms and phrases. Also, we added more details about CALAS and removed some redundant parts.
>
> ## Q1. How extracted causality used in forecasting?
>
> > Q1-A1. Even though we utilize CALAS in the input layer (as depicted in Fig. 3), the CALAS module could be utilized in any layer – input, output, or intermediate layers. We add an explanation for CALAS’s layer-agnostic usages.
>
>
> ## Q2. Please describe the ablation study setting in detail.
>
> > Q2-A1. We conduct ablation studies with an input length of 336 and 96, 192, 336, and 720 prediction lengths and a linear backbone, which means the backbone is a direct linear projection from input to output. In the ablation study, we replace the hypernetworks with original convolutional weights (i.e., static maps). The performances in the ablation study are reported on average over the performances over four different prediction lengths. We clarify the points in the manuscripts accordingly.
>
>
>
> [1] Judea Pearl, “Causality: Models, Reasoning, and Inference,” Cambridge University Press, New York. 2nd Edition, 2009
>
> [2] Judea Pearl, “Causal Inference in Statistics: An overview,” Statistics Surveys, 2009
>
> [3] Assaad et al., “Survey and evaluation of causal discovery methods for time series,” Journal of Artificial Intelligence Research, 2022

---

### Author Response · Authors · 2024-11-28
**Rebuttal Uploaded**

Dear Reviewers,

Thank you for the valuable and constructive comments. We greatly appreciate your insights, which have helped us improve our manuscripts.

We have uploaded our updated manuscripts in Openreview. We additionally uploaded the colored version of our manuscripts in the supplementary materials to facilitate easy tracking of the changes. Below, we summarize the main revisions:


1. We clarified vague terms and settings, removed redundant sentences, and improved the overall paper presentation.
2. We newly added the Section 4.1 for the discussion on causality, causal discovery and Granger causality to address the concerns raised about these topics.
3. In the appendix, we provide additional experimental results for two tasks: time series causal discovery and imputation. The experimental results on causal discovery demonstrate that CALAS effectively learns the causal relationship , even in the traditional causal discovery tasks. We are also in the process of updating the experimental results for anomaly detection and classification.
4. We have moved the discussion section into the appendix and included a discussion for CNN-, RNN-, and SSM-based approaches in modeling channel dependency. Additionally, we clarified the distinctions between CALAS and these approaches.


We will soon provide detailed answers for each reviewers.

Once again, thank you for your valuable feedback and the opportunity to improve our work.


Sincerely,
Authors

---

### Meta-Review · Area_Chair_qKa4 · 2024-12-17

**Metareview:**

The paper introduces CALAS, a deep learning model for multivariate time series forecasting that aims to capture causal relationships among variables by modeling causal strength and propagation delay. It claims improved forecasting accuracy over state-of-the-art methods.

Strengths include the novel approach to modeling causality in time series, potential for better interpretability, and demonstrated improvements over baseline models.

Weaknesses include a lack of clarity in defining "causality," insufficient distinction from correlation, vague language, and limited experimental evaluation. The paper also lacks open-source code, which hinders reproducibility.

The decision to reject is based on the paper's misuse of the term "causality," lack of precision in methodology, and the absence of code for reproducibility, despite some promising aspects in modeling causality in time series.

**Additional Comments On Reviewer Discussion:**

Reviewer 2T1p  and Reviewer x5dP think that, judging by the results, the proposed method merely models lag-based channel mixing, and presenting causality as the primary contribution could be misleading. The authors do not fully solve this problem.

---

### Decision · Program_Chairs · 2025-01-22

Reject